# Benchmarking Sustainable Mobility in Higher Education

Giulio Mario Cappelletti [1,*], Luca Grilli [1], Carlo Russo [1] and Domenico Santoro [2]

1   Department of Economics, Management and Territory, University of Foggia, 71121 Foggia, Italy;
    luca.grilli@unifg.it (L.G.); carlo.russo@unifg.it (C.R.)
2   Department of Economics and Finance, University of Bari Aldo Moro, 70124 Bari, Italy;
    domenico.santoro@uniba.it
*   Correspondence: giulio.cappelletti@unifg.it

**Abstract:** Sustainable mobility is an increasingly significant issue that both public and private organizations consider in order to reduce emissions by their members. In this paper, the Life Cycle Assessment (LCA) approach was used to evaluate sustainable mobility. Data coming from a study carried out at the University of Foggia were processed by Gabi LCA software to estimate the environmental performance of the community members according to the methodology of the Product Environmental Footprint (PEF) guidelines 3.0. Results of the LCA were organized in different classes, creating an eco-indicator of sustainable mobility that can be applied to both the institution and individual members (called the Sustainable Mobility Indicator, SMI). The SMI, computed to assess the environmental impact of the University of Foggia, was also used to evaluate the best mobility scenario, which can be considered a benchmark. The creation of the performance classes and benchmark analysis represents an easier way to communicate sustainability based on the recommendations for achieving the sustainable development goals from the 2030 Agenda adopted by all United Nations Member States. Indeed, any organization can carry out this approach to assess its environmental impact (in terms of mobility) and shape transport policies accordingly, leading to the adoption of sustainable solutions.

**Keywords:** Sustainable Mobility; SDG 11; Life Cycle Assessment; Sustainable Eco-Indicator University

## 1. Introduction

Sustainable mobility is a crucial issue for determining transport policy [1,2]. This is a global problem that concerns urban planning, all economic sectors, and higher education as well [3–6]. To assess sustainability in transport plans, many efforts have been carried out to determine metrics and indicators to face the problem in its three dimensions: economic (as for Moghaddam et al. [7], who compare inequality in transportation), social, and environmental [8–10]. Previous studies focused on the calculation of indexes based on the evaluation of various characteristics of sustainable mobility. Haghshenas and Vaziri [11] compared sustainable mobility indicators calculated on a global scale. On the other hand, Shiau and Liu [12] determined an indicator system for measuring and monitoring transport sustainability at the city level. In the same way, Jain and Tiwari [13] proposed a systematic approach to selecting sustainable mobility indicators for Indian cities. Mirzahossein et al. [14] investigate the traffic capacity under environmental constraints, calculating the maximum number of vehicles based on acceptable emission levels. Furthermore, adopting standardized methodologies to analyze transport modes and mobility plans from a life cycle perspective could help in defining an overall picture of sustainability and assessing the implications of choices and policies. Indeed, Life Cycle Assessment (LCA) has usually been adopted to evaluate the sustainability of mobility [15–20]. Starting from these premises, this paper proposes a way to calculate an indicator for assessing sustainable mobility in higher education. The paper analyses the results of a survey carried out in the academic community of the University of Foggia with the engagement of students,

professors, and technical staff [21,22]. Among the data collected, this paper focuses on the various kinds of transport modes and the kilometers traveled with each one. These data were used as life cycle inventory for calculating the environmental performance of each transport choice according to LCA methodology and PEF guidelines 3.0. The research objective was to individuate a metric for calculating an indicator based on the environmental impact associated with the choice of transport mode, while the expected results were to determine performance classes as an easier way to communicate sustainability based on the recommendations for achieving the sustainable development goals from the 2030 Agenda adopted by all United Nations Member States. This approach appears to be in line with other experiments in which sustainable mobility was assessed according to a scoring process calculated for several elements of the mobility plan [23]. Then, a further effort was made to elaborate a procedure for benchmarking the environmental performances calculated according to the sustainable mobility indicator. This represents an innovative aspect based on the concept of continual improvement indicated in the standards for quality management systems [24]. This approach forces the organization to compare its real situation with the best available solution from an environmental perspective and helps it manage sustainable mobility. Miranda and Rodrigues da Silva [25] used the same approach for benchmarking sustainable mobility in Curitiba (Brazil). Thus, the model proposed in this paper could be replicable in other academic communities or applicable to other organizations.

## 2. Materials and Methods

*Life Cycle Assessment of the Mobility of University of Foggia*

The LCA is a standardized methodology that aims to assess the environmental burdens of a product, service, or organization by considering the overall system in terms of material and energy resources consumption (input) and emissions (output) [26–33]. The LCA was applied to the two scenarios of the University of Foggia (UNIFG) mobility habits, distinguished between hot and cold seasons. This distinction was important because, according to the survey results, conditions could highly affect the choices of transport modes.

The modeling phase was carried out by using the LCA software Gabi by Sphera Solutions, and its data sets included Ecoinvent v3.5 [34,35]. Table 1 shows the processes and data sets considered in the system, distinguishing between Sphera and Ecoinvent. For each transportation mode, as indicated in Table 1, the impacts of fuel production and use, as well as use of vehicles, were included. According to LCA methodology, as for proxy data on processes of petrol, diesel, LPG, methane, and electric cars, deriving from Ecoinvent and Sphera data sets, the functional unit was the kilometer. The same was true about scooters. On the other hand, as far as trains, buses, and aircraft, all impacts are referred to by the unit "passenger kilometers" (pkm) [36]. This choice is based on the need to consider that the impacts of public transport must be divided per the average capacity of the vehicles in terms of carried persons. As for sharing mobility, a multiplying factor of 0.25 was applied to the impact of passenger cars, whereas for hybrid vehicles, 80% of petrol cars and 20% of electric cars were considered. As far as the life cycle impact assessment, all the impact categories indicated in the Product Environmental Footprint (PEF) guidelines 3.0 were considered [37,38]. The results in absolute value were normalized and weighted according to Table 2 in order to obtain an aggregate indicator named "EF 3.0 eco indicator".

**Table 1.** Processes and data sets considered in the LCA.

| Process | Dataset |
|---|---|
| Car diesel, small size Euro 0 | EU-28: Diesel mix at refinery Sphera |
| | GLO: Car diesel, 1986–88, engine size up to 1.4l Sphera |
| Car diesel, small size Euro 1 | EU-28: Diesel mix at refinery Sphera |
| | GLO: Car diesel, Euro 1, engine size up to 1.4l Sphera |
| Car diesel, small size Euro 2 | EU-28: Diesel mix at refinery Sphera |
| | GLO: Car diesel, Euro 2, engine size up to 1.4l Sphera |
| Car diesel, small size Euro 3 | EU-28: Diesel mix at refinery Sphera |
| | GLO: Car diesel, Euro 3, engine size up to 1.4l Sphera |
| Car diesel, small size Euro 4 | EU-28: Diesel mix at refinery Sphera |
| | GLO: Car diesel, Euro 4, engine size up to 1.4l Sphera |
| Car diesel, small size Euro 5 | EU-28: Diesel mix at refinery Sphera |
| | GLO: Car diesel, Euro 5, engine size up to 1.4l Sphera |
| Car diesel, small size Euro 6 | EU-28: Diesel mix at refinery Sphera |
| | GLO: Car diesel, Euro 6, engine size up to 1.4l Sphera |
| Car diesel, small size Euro 6b | EU-28: Diesel mix at refinery Sphera |
| | GLO: Car diesel, Euro 6 (from Sept 2019), engine size up to 1.4l Sphera |
| Car diesel, small size Euro 6c | EU-28: Diesel mix at refinery Sphera |
| | GLO: Car diesel, Euro 6 (from January 2021), engine size up to 1.4l Sphera |
| Car diesel, medium size Euro 0 | EU-28: Diesel mix at refinery Sphera |
| | GLO: Car diesel, 1986–88, engine size 1.4–2l Sphera |
| Car diesel, medium size Euro 1 | EU-28: Diesel mix at refinery Sphera |
| | GLO: Car diesel, Euro 1, engine size 1.4–2l Sphera |
| Car diesel, medium size Euro 2 | EU-28: Diesel mix at refinery Sphera |
| | GLO: Car diesel, Euro 2, engine size 1.4–2l Sphera |
| Car diesel, medium size Euro 3 | EU-28: Diesel mix at refinery Sphera |
| | GLO: Car diesel, Euro 3, engine size 1.4–2l Sphera |
| Car diesel, medium size Euro 4 | EU-28: Diesel mix at refinery Sphera |
| | GLO: Car diesel, Euro 4, engine size 1.4–2l Sphera |
| Car diesel, medium size Euro 5 | EU-28: Diesel mix at refinery Sphera |
| | GLO: Car diesel, Euro 5, engine size 1.4–2l Sphera |
| Car diesel, medium size Euro 6 | EU-28: Diesel mix at refinery Sphera |
| | GLO: Car diesel, Euro 6, engine size 1.4–2l Sphera |
| Car diesel, medium size Euro 6b | EU-28: Diesel mix at refinery Sphera |
| | GLO: Car diesel, Euro 6 (from September 2019), engine size 1.4–2l Sphera |
| Car diesel, medium size Euro 6c | EU-28: Diesel mix at refinery Sphera |
| | GLO: Car diesel, Euro 6 (from January 2021), engine size 1.4–2l Sphera |
| Car diesel, large size Euro 0 | EU-28: Diesel mix at refinery Sphera |
| | GLO: Car diesel, 1986-88, engine size more than 2l Sphera |
| Car diesel, large size Euro 1 | EU-28: Diesel mix at refinery Sphera |
| | GLO: Car diesel, Euro 1, engine size more than 2l Sphera |

**Table 1.** *Cont.*

| Process | Dataset |
|---|---|
| Car diesel, large size Euro 2 | EU-28: Diesel mix at refinery Sphera |
| | GLO: Car diesel, Euro 2, engine size more than 2l Sphera |
| Car diesel, large size Euro 3 | EU-28: Diesel mix at refinery Sphera |
| | GLO: Car diesel, Euro 3, engine size more than 2l Sphera |
| Car diesel, large size Euro 4 | EU-28: Diesel mix at refinery Sphera |
| | GLO: Car diesel, Euro 4, engine size more than 2l Sphera |
| Car diesel, large size Euro 5 | EU-28: Diesel mix at refinery Sphera |
| | GLO: Car diesel, Euro 5, engine size more than 2l Sphera |
| Car diesel, large size Euro 6 | EU-28: Diesel mix at refinery Sphera |
| | GLO: Car diesel, Euro 6, engine size more than 2l Sphera |
| Car diesel, large size Euro 6b | EU-28: Diesel mix at refinery Sphera |
| | GLO: Car diesel, Euro 6 (from September 2019), engine size more than 2l Sphera |
| Car diesel, large size Euro 6c | EU-28: Diesel mix at refinery Sphera |
| | GLO: Car diesel, Euro 6 (from January 2021), engine size more than 2l Sphera |
| Car petrol, small size Euro 0 | EU-28: Gasoline mix (regular) at refinery Sphera |
| | GLO: Car petrol, controlled catalytic converter 87–90, engine size up to 1.4l Sphera |
| Car petrol, small size Euro 1 | EU-28: Gasoline mix (regular) at refinery Sphera |
| | GLO: Car petrol, Euro 1, engine size up to 1.4l Sphera |
| Car petrol, small size Euro 2 | EU-28: Gasoline mix (regular) at refinery Sphera |
| | GLO: Car petrol, Euro 2, engine size up to 1.4l Sphera |
| Car petrol, small size Euro 3 | EU-28: Gasoline mix (regular) at refinery Sphera |
| | GLO: Car petrol, Euro 3, engine size up to 1.4l Sphera |
| Car petrol, small size Euro 4 | EU-28: Gasoline mix (regular) at refinery Sphera |
| | GLO: Car petrol, Euro 4, engine size up to 1.4l Sphera |
| Car petrol, small size Euro 5 | EU-28: Gasoline mix (regular) at refinery Sphera |
| | GLO: Car petrol, Euro 5, engine size up to 1.4l Sphera |
| Car petrol, small size Euro 6 | EU-28: Gasoline mix (regular) at refinery Sphera |
| | GLO: Car petrol, Euro 6, engine size up to 1.4l Sphera |
| Car petrol, medium size Euro 0 | EU-28: Gasoline mix (regular) at refinery Sphera |
| | GLO: Car petrol, controlled catalytic converter 87–90, engine size 1.4-2l Sphera |
| Car petrol, medium size Euro 1 | EU-28: Gasoline mix (regular) at refinery Sphera |
| | GLO: Car petrol, Euro 1, engine size 1.4–2l Sphera |
| Car petrol, medium size Euro 2 | EU-28: Gasoline mix (regular) at refinery Sphera |
| | GLO: Car petrol, Euro 2, engine size 1.4–2l Sphera |
| Car petrol, medium size Euro 3 | EU-28: Gasoline mix (regular) at refinery Sphera |
| | GLO: Car petrol, Euro 3, engine size 1.4-2l Sphera |
| Car petrol, medium size Euro 4 | EU-28: Gasoline mix (regular) at refinery Sphera |
| | GLO: Car petrol, Euro 4, engine size 1.4-2l Sphera |
| Car petrol, medium size Euro 5 | EU-28: Gasoline mix (regular) at refinery Sphera |
| | GLO: Car petrol, Euro 5, engine size 1.4-2l Sphera |

**Table 1.** *Cont.*

| Process | Dataset |
|---|---|
| Car petrol, medium size Euro 6 | EU-28: Gasoline mix (regular) at refinery Sphera |
| | GLO: Car petrol, Euro 6, engine size 1.4-2l Sphera |
| Car petrol, large size Euro 0 | EU-28: Gasoline mix (regular) at refinery Sphera |
| | GLO: Car petrol, controlled catalytic converter 87–90, engine size more than 2l Sphera |
| Car petrol, large size Euro 1 | EU-28: Gasoline mix (regular) at refinery Sphera |
| | GLO: Car petrol, Euro 1, engine size more than 2l Sphera |
| Car petrol, large size Euro 2 | EU-28: Gasoline mix (regular) at refinery Sphera |
| | GLO: Car petrol, Euro 2, engine size more than 2l Sphera |
| Car petrol, large size Euro 3 | EU-28: Gasoline mix (regular) at refinery Sphera |
| | GLO: Car petrol, Euro 3, engine size more than 2l Sphera |
| Car petrol, large size Euro 4 | EU-28: Gasoline mix (regular) at refinery Sphera |
| | GLO: Car petrol, Euro 4, engine size more than 2l Sphera |
| Car petrol, large size Euro 5 | EU-28: Gasoline mix (regular) at refinery Sphera |
| | GLO: Car petrol, Euro 5, engine size more than 2l Sphera |
| Car petrol, large size Euro 6 | EU-28: Gasoline mix (regular) at refinery Sphera |
| | GLO: Car petrol, Euro 6, engine size more than 2l Sphera |
| Car Methane | DE: Methane Sphera |
| | GLO: Car CNG, Euro 3 Sphera |
| Car LPG | GLO: Car LPG, Euro 3 Sphera |
| Car Electric | GLO: market for transport, passenger car, electric Ecoinvent 3.5 |
| Car Hybrid | GLO: market for transport, passenger car, electric Ecoinvent 3.5 |
| | GLO: Passenger car, average, Euro 3-5, engine size from 1.4l up to >2l Sphera |
| Scooter | GLO: market for transport, passenger, motor scooter Ecoinvent 3.5 |
| BUS | GLO: market for transport, regular bus Ecoinvent 3.5 |
| Train | IT: transport, passenger train Ecoinvent 3.5 |
| Sharing Mobility | GLO: Passenger car, average, Euro 3-5, engine size from 1.4l up to >2l Sphera |
| Aircraft | GLO: market for transport, passenger, aircraft Ecoinvent 3.5 |

**Table 2.** EF 3.0 normalization factors (person equivalents) and weighting factors.

| Impact Category | Unit | Normalization Factors (Person Equivalents) | Weighting Factors |
|---|---|---|---|
| EF 3.0 Acidification | Mole of H+ Equation | 0.017986 | 6.2 |
| EF 3.0 Climate change—total | kg CO2 Equation | 0.000124 | 21.06 |
| EF 3.0 Ecotoxicity, freshwater–total | CTUe | $2.34 \times 10^{-5}$ | 1.92 |
| EF 3.0 Eutrophication, freshwater | kg P Equation | 0.621118 | 2.8 |
| EF 3.0 Eutrophication, marine | kg N Equation | 0.051282 | 2.96 |
| EF 3.0 Eutrophication, terrestrial | Mole of N Equation | 0.00565 | 3.71 |
| EF 3.0 Human toxicity, cancer—total | CTUh | 53763.44 | 2.13 |
| EF 3.0 Human toxicity, non-cancer—total | CTUh | 4347.826 | 1.84 |
| EF 3.0 Ionising radiation, human health | kBq U235 Equation | 0.007246 | 5.01 |

**Table 2.** *Cont.*

| Impact Category | Unit | Normalization Factors (Person Equivalents) | Weighting Factors |
|---|---|---|---|
| EF 3.0 Land use | Pt | $4.48 \times 10^{-7}$ | 7.94 |
| EF 3.0 Ozone depletion | kg CFC-11 Equation | 20.66116 | 6.31 |
| EF 3.0 Particulate matter | Disease Incidences | 1680.672 | 8.96 |
| EF 3.0 Photochemical ozone formation, human health | kg NMVOC Equation | 0.02457 | 4.78 |
| EF 3.0 Resource use, fossils | MJ | $1.54 \times 10^{-5}$ | 8.32 |
| EF 3.0 Resource use, mineral and metals | kg Sb Equation | 15.72327 | 7.55 |
| EF 3.0 Water use | $m^3$ World Equiv. | $8.70 \times 10^{-5}$ | 8.51 |

## 3. Results

### 3.1. Analysis of the EF 3.0 eco-Indicator for Transport Modes

In Table 3, for each transport mode, and distinguishing between hot and cold seasons, kilometers are compared with the EF 3.0 eco-indicator calculated, multiplying the former by the relative impact per kilometer. It is essential to point out that the contribution of fuel production is about 15% in the case of diesel cars and over 20% for petrol cars; the rest of the impacts refer to the other phases of the life cycle. At the same time, it is worth noting that a large-sized Euro 5 diesel car presents a value a little higher than that of the same Euro 4 vehicles (around 5% more).

**Table 3.** Kilometers vs. EF 3.0 eco-indicator for both the hot and cold seasons.

| | km | | | EF 3.0 Eco-Indicator | |
|---|---|---|---|---|---|
| | Hot Season | Cold Season | per km | Hot Season | Cold Season |
| Car diesel, small size Euro 0 | 0 | 1320 | $1.29 \times 10^{-3}$ | 0.00 | 1.71 |
| Car diesel, small size Euro 1 | 0 | 66 | $1.31 \times 10^{-3}$ | 0.00 | 0.09 |
| Car diesel, small size Euro 2 | 752 | 1328 | $1.14 \times 10^{-3}$ | 0.85 | 1.51 |
| Car diesel, small size Euro 3 | 44,658 | 67,670 | $1.03 \times 10^{-3}$ | 46.13 | 69.90 |
| Car diesel, small size Euro 4 | 58,984 | 105,345 | $9.39 \times 10^{-4}$ | 55.39 | 98.92 |
| Car diesel, small size Euro 5 | 5500 | 18,827 | $1.02 \times 10^{-3}$ | 5.59 | 19.15 |
| Car diesel, small size Euro 6 | 15,937 | 12,891 | $8.20 \times 10^{-4}$ | 13.07 | 10.58 |
| Car diesel, small size Euro 6b | 2240 | 4662 | $6.69 \times 10^{-4}$ | 1.50 | 3.12 |
| Car diesel, small size Euro 6c | 12,888 | 22,758 | $6.44 \times 10^{-4}$ | 8.31 | 14.67 |
| Car diesel, medium size Euro 0 | 0 | 0 | $1.71 \times 10^{-3}$ | 0.00 | 0.00 |
| Car diesel, medium size Euro 1 | 4260 | 0 | $1.71 \times 10^{-3}$ | 7.27 | 0.00 |
| Car diesel, medium size Euro 2 | 15,698 | 34,581 | $1.50 \times 10^{-3}$ | 23.60 | 51.98 |
| Car diesel, medium size Euro 3 | 131,916 | 239,685 | $1.31 \times 10^{-3}$ | 172.51 | 313.44 |
| Car diesel, medium size Euro 4 | 199,252 | 372,083 | $1.13 \times 10^{-3}$ | 225.97 | 421.98 |
| Car diesel, medium size Euro 5 | 154,650 | 283,778 | $1.20 \times 10^{-3}$ | 186.34 | 341.93 |
| Car diesel, medium size Euro 6 | 121,688 | 234,502 | $9.99 \times 10^{-4}$ | 121.52 | 234.17 |
| Car diesel, medium size Euro 6b | 50,696 | 84,202 | $8.47 \times 10^{-4}$ | 42.93 | 71.31 |

**Table 3.** *Cont.*

| | km | | EF 3.0 Eco-Indicator | | |
|---|---|---|---|---|---|
| | **Hot Season** | **Cold Season** | **per km** | **Hot Season** | **Cold Season** |
| Car diesel, medium size Euro 6c | 90,631 | 161,111 | $8.23 \times 10^{-4}$ | 74.55 | 132.53 |
| Car diesel, large size Euro 0 | 0 | 0 | $2.07 \times 10^{-3}$ | 0.00 | 0.00 |
| Car diesel, large size Euro 1 | 0 | 0 | $2.06 \times 10^{-3}$ | 0.00 | 0.00 |
| Car diesel, large size Euro 2 | 0 | 0 | $1.82 \times 10^{-3}$ | 0.00 | 0.00 |
| Car diesel, large size Euro 3 | 7596 | 14,564 | $1.58 \times 10^{-3}$ | 12.00 | 23.00 |
| Car diesel, large size Euro 4 | 14,787 | 30,571 | $1.44 \times 10^{-3}$ | 21.26 | 43.94 |
| Car diesel, large size Euro 5 | 13,992 | 18,568 | $1.47 \times 10^{-3}$ | 20.53 | 27.25 |
| Car diesel, large size Euro 6 | 2728 | 6208 | $1.23 \times 10^{-3}$ | 3.36 | 7.65 |
| Car diesel, large size Euro 6b | 6240 | 10,920 | $1.08 \times 10^{-3}$ | 6.74 | 11.79 |
| Car diesel, large size Euro 6c | 1710 | 3867 | $1.06 \times 10^{-3}$ | 1.81 | 4.08 |
| Car petrol, small size Euro 0 | 0 | 83 | $1.46 \times 10^{-3}$ | 0.00 | 0.12 |
| Car petrol, small size Euro 1 | 2971 | 13,674 | $1.43 \times 10^{-3}$ | 4.24 | 19.52 |
| Car petrol, small size Euro 2 | 5968 | 12,974 | $1.28 \times 10^{-3}$ | 7.65 | 16.64 |
| Car petrol, small size Euro 3 | 28,050 | 55,630 | $1.06 \times 10^{-3}$ | 29.73 | 58.97 |
| Car petrol, small size Euro 4 | 70,413 | 142,469 | $1.00 \times 10^{-3}$ | 70.71 | 143.07 |
| Car petrol, small size Euro 5 | 21,117 | 38,547 | $9.51 \times 10^{-4}$ | 20.08 | 36.66 |
| Car petrol, small size Euro 6 | 26,970 | 59,364 | $9.15 \times 10^{-4}$ | 24.69 | 54.34 |
| Car petrol, small size Euro 6b | 7227 | 11,670 | $9.15 \times 10^{-4}$ | 6.62 | 10.68 |
| Car petrol, small size Euro 6c | 9389 | 22,584 | $9.15 \times 10^{-4}$ | 8.60 | 20.67 |
| Car petrol, medium size Euro 0 | 0 | 0 | $1.79 \times 10^{-3}$ | 0.00 | 0.00 |
| Car petrol, medium size Euro 1 | 0 | 0 | $1.71 \times 10^{-3}$ | 0.00 | 0.00 |
| Car petrol, medium size Euro 2 | 3784 | 6718 | $1.55 \times 10^{-3}$ | 5.85 | 10.39 |
| Car petrol, medium size Euro 3 | 6696 | 18,953 | $1.30 \times 10^{-3}$ | 8.67 | 24.55 |
| Car petrol, medium size Euro 4 | 24,232 | 35,049 | $1.19 \times 10^{-3}$ | 28.93 | 41.84 |
| Car petrol, medium size Euro 5 | 12,502 | 27,028 | $1.13 \times 10^{-3}$ | 14.15 | 30.59 |
| Car petrol, medium size Euro 6 | 7830 | 10,880 | $1.08 \times 10^{-3}$ | 8.45 | 11.74 |
| Car petrol, medium size Euro 6b | 19,983 | 48,143 | $1.08 \times 10^{-3}$ | 21.56 | 51.95 |
| Car petrol, medium size Euro 6c | 7261 | 15,131 | $1.08 \times 10^{-3}$ | 7.84 | 16.33 |
| Car petrol, large size Euro 0 | 0 | 0 | $2.21 \times 10^{-3}$ | 0.00 | 0.00 |
| Car petrol, large size Euro 1 | 0 | 0 | $2.15 \times 10^{-3}$ | 0.00 | 0.00 |
| Car petrol, large size Euro 2 | 0 | 0 | $1.97 \times 10^{-3}$ | 0.00 | 0.00 |
| Car petrol, large size Euro 3 | 0 | 0 | $1.71 \times 10^{-3}$ | 0.00 | 0.00 |
| Car petrol, large size Euro 4 | 636 | 1254 | $1.65 \times 10^{-3}$ | 1.05 | 2.06 |
| Car petrol, large size Euro 5 | 0 | 0 | $1.58 \times 10^{-3}$ | 0.00 | 0.00 |
| Car petrol, large size Euro 6 | 57 | 99 | $1.53 \times 10^{-3}$ | 0.09 | 0.15 |
| Car petrol, large size Euro 6b | 0 | 0 | $1.53 \times 10^{-3}$ | 0.00 | 0.00 |

**Table 3.** *Cont.*

| | km | | EF 3.0 Eco-Indicator | | |
|---|---|---|---|---|---|
| | **Hot Season** | **Cold Season** | **per km** | **Hot Season** | **Cold Season** |
| Car petrol, large size Euro 6c | 0 | 0 | $1.53 \times 10^{-3}$ | 0.00 | 0.00 |
| Car Methane | 91,705 | 179,936 | $1.13 \times 10^{-3}$ | 103.23 | 202.56 |
| Car LPG | 136,954 | 259,970 | $1.19 \times 10^{-3}$ | 162.81 | 309.06 |
| Car Electric | 0 | 0 | $4.36 \times 10^{-3}$ | 0.00 | 0.00 |
| Car Hybrid | 13,901 | 36,493 | $1.84 \times 10^{-3}$ | 25.57 | 67.11 |
| Scooter | 1087 | 6091 | $1.54 \times 10^{-3}$ | 1.68 | 9.39 |
| BUS | 1,640,107 | 3,723,199 | $1.42 \times 10^{-3}$ | 2334.35 | 5299.19 |
| Train | 2,796,495 | 5,928,588 | $7.44 \times 10^{-4}$ | 2080.27 | 4410.18 |
| Sharing Mobility | 329 | 828 | $3.02 \times 10^{-4}$ | 0.10 | 0.25 |
| Aircraft | 27,000 | 80,700 | $1.22 \times 10^{-3}$ | 33.04 | 98.75 |

The same situation is highlighted for medium-sized diesel cars, Euro 5 to Euro 4 cars, and for small-size diesel cars, Euro 4 to Euro 5. Furthermore, Table 3 shows, concerning electric or hybrid vehicles and methane cars, how their performances are not so sustainable, which is in line with other previous studies. This is principally due to the technological aspects linked to production and disposal, especially for batteries, and also the prevalence of fossil fuels in the power mix [35,38–46]. The accuracy of these results derives from the use of proxy data of the Sphera and Ecoinvent data sets used in the analysis, so we can assert that according to the results shown in Table 3, sharing mobility and public transport remain the most sustainable choices from a life cycle perspective.

*3.2. The Environmental Performances of Mobility at the University of Foggia*

The results in Table 3 are summarized in Figures 1 and 2, in which relative contributions of the primary transport mode are compared with the relative kilometers for the hot and cold seasons, respectively. The train appears attractive because while it represents over 47% of the total kilometers, its contribution in terms of impact is only almost 35%. On the other hand, as for the bus, its contribution of nearly 30% to the total kilometers becomes over 38% if its relative contribution is translated in terms of emissions. In the same way, the contribution of diesel cars changes from almost 14% in kilometers to over 17% concerning the eco-indicator. Regarding petrol cars and other transport modes, the percentage of the total impact does not change significantly concerning kilometers. As in the hot season, the train and bus covered around 75% of the total kilometers traveled in the cold season, and their advantages in terms of environmental performance, especially regarding the use of trains, are highlighted in Figure 2. The relationship between impact and kilometers of diesel and petrol cars appears slightly lower than that of the hot season due to the increase in the use of small cars. This is not true for the other transport modes, for which the same relationship highlighted for the hot season is detected.

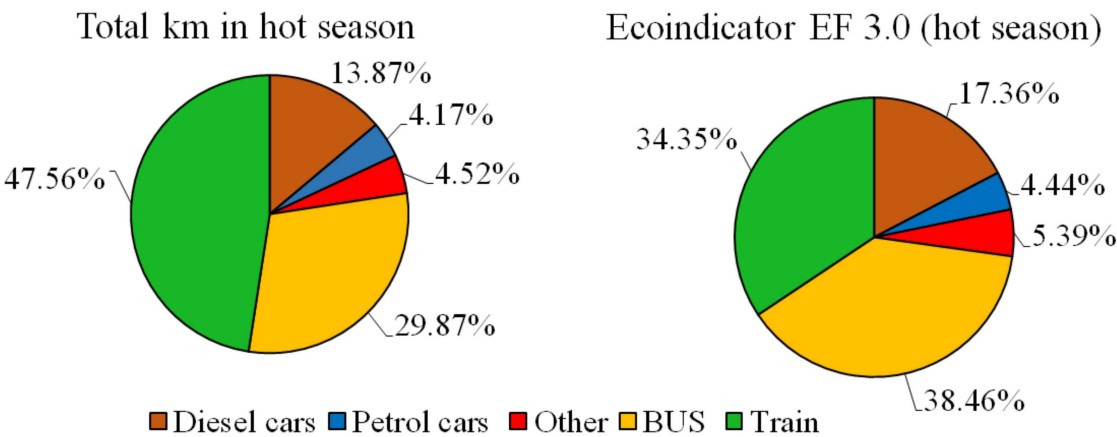

**Figure 1.** Contribution of the main transport mode in the hot season.

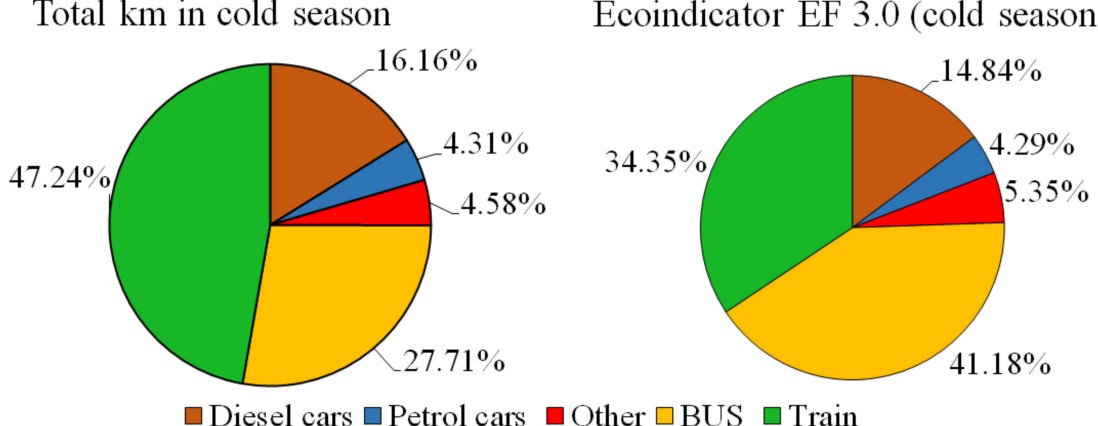

**Figure 2.** Contribution of the main transport mode in the cold season.

*3.3. Benchmarking Sustainable Mobility in Higher Education*

Starting from this analysis and considering the elaboration of the information collected through the survey at the University of Foggia, it is possible to formulate a simple indicator that is easy for all stakeholders in higher education to understand. The Sustainable Mobility Indicator (*SMI*) aims to express by a non-dimensional number the environmental performance class of the overall community. The value is calculated according to Equation (1).

$$SMI = \frac{\sum_{i=1}^{n}(kmi \times Ei)}{\sum_{i=1}^{n}kmi} \tag{1}$$

where:

- *kmi* represents the number of kilometers traveled using each transport mode, respectively, and for a certain period (year, season, week, etc.);
- *Ei* is the eco-indicator (in our case the EF 3.0 eco-indicator) calculated for the relative transport mode.

The SMI calculated according to Equation (1) could be compared with the best environmental performance deriving from the adoption of the best transport solution for all kilometers traveled. This latter is, in fact, the benchmark, and as the SMI negatively deviates from it, the performance class becomes worse. Table 4 shows the hypothesis of five performance classes calculated by multiplying the benchmark per 2, 4, 6, and 8, respectively. In the case of the University of Foggia, the situation is described in Figure 3. The performance class appears good. Indeed, the SMI is located in the first range.

**Table 4.** Hypothesis of performance classes.

| Performance Classes | Range | SMI | SMI UNIFG (Hot Season) | SMI UNIFG (Cold Season) |
|---|---|---|---|---|
| A | From Benchmark to Benchmark ×2 | From $7.44 \times 10^{-4}$ to $1.49 \times 10^{-3}$ | $1.02 \times 10^{-3}$ | $1.03 \times 10^{-3}$ |
| B | From Benchmark ×2 to Benchmark ×4 | From $1.50 \times 10^{-3}$ to $2.98 \times 10^{-3}$ | | |
| C | From Benchmark ×4 to Benchmark ×6 | From $2.99 \times 10^{-3}$ to $4.46 \times 10^{-3}$ | | |
| D | From Benchmark ×6 to Benchmark ×8 | From $4.47 \times 10^{-3}$ to $5.95 \times 10^{-3}$ | | |
| E | From Benchmark ×8 to Benchmark ×10 | Over $5.95 \times 10^{-3}$ | | |

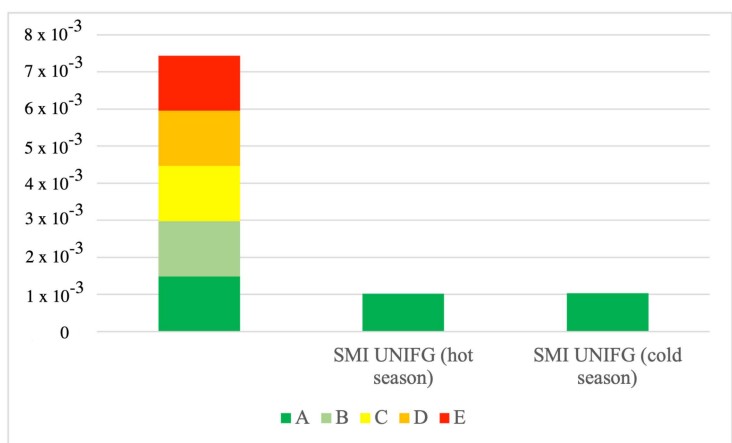

**Figure 3.** Comparison of the SMI between different performance classes.

## 4. Conclusions

The approach proposed in this paper for benchmarking sustainable mobility in higher education is based on the information collected by the use of a survey, as well as the environmental impact associated with the choice of transport mobility. It aims to elaborate performance classes based on a standardized methodology and communicate in an easier way the sustainability of transport modes by using the SMI. To enhance the model, it could be useful to stratify the sample and direct further analysis toward the attribution of an environmental profile for each component of the academic community. Despite limits and constraints linked to a large amount of data and information needed, this could play a crucial role in assessing the effects of mobility choices and evaluating their environmental implications.

This information could be very useful in managing mobility policies and addressing the sustainable habits of all community members. Further interesting analysis could be, for example, focused on the consequences of distance learning from a life cycle perspective. The advantages deriving from the lack of travel should be compared with the increasing use of energy (used for servers, computers, and electronic devices). In this way, the environmental profile determined by the SMI could be enriched with additional elements calculated according to LCA. Furthermore, the benchmarking phase in this paper is represented by the best situation for the particular organization, which could be referred to as an average performance identified by considering some specific parameters (e.g., geographical context, level of public investments in sustainable mobility of infrastructures). In the future, a certification system could be considered, and guidelines based on the approach proposed

in this paper could represent a milestone in assessing sustainability in higher education and encourage sustainable choice in transport mode.

**Author Contributions:** Conceptualization, G.M.C., L.G., C.R. and D.S.; methodology, G.M.C., L.G., C.R. and D.S.; validation, L.G., C.R. and D.S.; formal analysis, G.M.C., L.G., C.R. and D.S.; investigation, G.M.C., L.G., C.R. and D.S.; resources, G.M.C., L.G., C.R. and D.S.; data curation, L.G. and D.S.; writing—original draft preparation, G.M.C., L.G., C.R. and D.S.; writing—review and editing, G.M.C., L.G., C.R. and D.S.; visualization, L.G., C.R. and D.S.; supervision, G.M.C., L.G., C.R. and D.S. All authors have read and agreed to the published version of the manuscript. Authors are listed in alphabetic order.

**Funding:** This research received no external funding.

**Informed Consent Statement:** Informed consent was obtained from all subjects involved in the study.

**Data Availability Statement:** Data available on request; please contact the corresponding author (G.M.C.).

**Acknowledgments:** We want to thank Agostino Sevi, of the University of Foggia for his concrete support and encouragement to carry out this research.

**Conflicts of Interest:** The authors declare no conflict of interest.

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
