# Peer review of "Benchmarking Sustainable Mobility in Higher Education"

_sustainability, doi:10.3390/su15065190_

Round 1

Reviewer 1 Report

1. The results and discussions need more justification to be done.

2. methodology is not clear.

3. Problem statement is not clear.

4. The conclusions should be modified with justifications for the results.

5. Need to add recommendations. 

Author Response

  1. The results and discussions need more justification to be done.
  2. methodology is not clear.
  3. Problem statement is not clear.
  4. The conclusions should be modified with justifications for the results.
  5. Need to add recommendations. 

Answer:

Dear Reviewer

We would like to thank for your comments. We rearranged the text according to your suggestion. In particular:

Results and discussion were enhanced

Methodology is well explained both in the abstract and introduction and in the section Materials and methods

Problem statement is clarified

Conclusions were enhanced

Recommendations were added

Reviewer 2 Report

The study focuses on analyzing the environmental impact of the transportation modes preferred by the member of the University of Foggia using the obtained survey results. The methodology and the contribution of the study are clearly presented. I only have a couple of minor comments. Although there is an independent study about the survey, there is no explanation about the survey in the current paper. Adding a paragraph that presents the basic properties of the survey (i.e., the number of participants, the number of questions, etc.) can help to improve the consistency. Besides, there are plenty of typos in the manuscript such as redundant numbers and unnecessary “-” symbols that separates a word (i.e., pa-per, car-ried, en-gagenent, elabo-rate,…). Please proofread the manuscript to correct all types of typos.     

Author Response

Reviewer 2:

The study focuses on analyzing the environmental impact of the transportation modes preferred by the member of the University of Foggia using the obtained survey results. The methodology and the contribution of the study are clearly presented. I only have a couple of minor comments. Although there is an independent study about the survey, there is no explanation about the survey in the current paper. Adding a paragraph that presents the basic properties of the survey (i.e., the number of participants, the number of questions, etc.) can help to improve the consistency. Besides, there are plenty of typos in the manuscript such as redundant numbers and unnecessary “-” symbols that separates a word (i.e., pa-per, car-ried, en-gagenent, elabo-rate,…). Please proofread the manuscript to correct all types of typos.

Answer:

Dear Reviewer

We would like to thank for your comments. As for the survey, any information is available on the following references cited in the text. In particular we added another reference recently published.

Cappelletti, G.M.; Grilli, L.; Russo, C.; Santoro, D. Sustainable Mobility in Universities: The Case of the University of Foggia (Italy). Environments 2021, 8. https://doi.org/10.3390/environments8060057

Cappelletti, G.M.; Grilli, L.; Russo, C.; Santoro, D. Machine Learning and Sustainable Mobility: The Case of the University of Foggia (Italy). Appl. Sci. 2022, 12, 8774. https://doi.org/10.3390/app12178774

Reviewer 3 Report

The paper mentioned In order to encourage their members to produce fewer greenhouse gas emissions, public and commercial organizations are increasingly giving thought to the need to promote environmentally friendly modes of transportation. In order to have a technique that is standardized and appropriate for all circumstances, the Life Cycle Assessment (LCA) is typically utilized to do an evaluation of sustainable mobility. It was possible to estimate the emissions of the community members through the creation of different classes in this paper, which was based on the study of sustainable mobility that was carried out at the University of Foggia, as the authors mention. Although it is not appropriate to mention any references in the abstract since the abstract should contain the author's contributions and try to sum up the whole of the manuscript. Moreover, they suggested the SMI, which was computed to analyze the University of Foggia's influence on the environment and was also used to evaluate the optimal mobility scenario, which may be regarded as a benchmark. This evaluation was done using the SMI. Nevertheless, any firm may do this kind of scenario analysis in order to evaluate its impact on the environment (in terms of mobility), with the goal of influencing transportation legislation that will ultimately result in the adoption of sustainable solutions.
Although it may be great work, it is not in the form of a research paper, especially the writing. For example, the methodology is unclear, as is the contribution.

That is one reason why I should mention that the paper is not well organized, and it could be considered a technical note and not a research paper.     

Best, Hamid 

Author Response

Reviewer 3:

The paper mentioned In order to encourage their members to produce fewer greenhouse gas emissions, public and commercial organizations are increasingly giving thought to the need to promote environmentally friendly modes of transportation. In order to have a technique that is standardized and appropriate for all circumstances, the Life Cycle Assessment (LCA) is typically utilized to do an evaluation of sustainable mobility. It was possible to estimate the emissions of the community members through the creation of different classes in this paper, which was based on the study of sustainable mobility that was carried out at the University of Foggia, as the authors mention. Although it is not appropriate to mention any references in the abstract since the abstract should contain the author's contributions and try to sum up the whole of the manuscript. Moreover, they suggested the SMI, which was computed to analyze the University of Foggia's influence on the environment and was also used to evaluate the optimal mobility scenario, which may be regarded as a benchmark. This evaluation was done using the SMI. Nevertheless, any firm may do this kind of scenario analysis in order to evaluate its impact on the environment (in terms of mobility), with the goal of influencing transportation legislation that will ultimately result in the adoption of sustainable solutions.
Although it may be great work, it is not in the form of a research paper, especially the writing. For example, the methodology is unclear, as is the contribution.

That is one reason why I should mention that the paper is not well organized, and it could be considered a technical note and not a research paper.     

Best, Hamid

Answer:

Dear Reviewer

We would like to thank for your comments. We took off the reference in the abstract. We also rearranged the text in order to make it suitable for publication.

Reviewer 4 Report

Thank you for submitting your paper to sustainability. The quality of writing is not acceptable. For instance, I would suggest reading the abstract once more. THe length of the manuscript is suitable for a conference paper, not a journal paper. I wish hope you improve your manuscript significantly for further submissions.

Author Response

Reviewer 4:

Thank you for submitting your paper to sustainability. The quality of writing is not acceptable. For instance, I would suggest reading the abstract once more. THe length of the manuscript is suitable for a conference paper, not a journal paper. I wish hope you improve your manuscript significantly for further submissions.

Dear Reviewer

We would like to thank for your comments. We rearranged the text in order to make it suitable for publication

Reviewer 5 Report

The manuscript has been analyzed with very interesting questions. The topic is original and addresses a specific gap in the field of sustainable mobility in higher education. The manuscript has to be improved in the abstract, introduction, and conclusion. In the abstract and introduction, it has to add the subject, research objectives, hypotheses, and expected results. The conclusions are consistent with the evidence and arguments presented. But, in conclusion, it has to add the main contribution and limitations of the study. The references are appropriate. The tables and figures are correct and are consistent with the results presented. 

Author Response

Reviewer 5:

The manuscript has been analyzed with very interesting questions. The topic is original and addresses a specific gap in the field of sustainable mobility in higher education. The manuscript has to be improved in the abstract, introduction, and conclusion. In the abstract and introduction, it has to add the subject, research objectives, hypotheses, and expected results. The conclusions are consistent with the evidence and arguments presented. But, in conclusion, it has to add the main contribution and limitations of the study. The references are appropriate. The tables and figures are correct and are consistent with the results presented.

Dear Reviewer

We would like to thank for your comments. We rearranged the sections: abstract, introduction and conclusion according to your suggestions.

Round 2

Reviewer 3 Report

  The paper has improved now, and it could be considered for publication after minor revisions, as it needs to consider the capacity of transportation with regard to environmental issues, as mentioned in the following paper as key role of sustainable transportation in the development, and enrich the scope of the paper since transportation is one of the main sources of sustainable cities. Moreover, economic issues as a one of the most important legs of sustainability would be considered in benchmarking. In this regard, it would be necessary to address this issue even in the upcoming proposed literature. https://doi.org/10.3390/su142114462 https://doi.org/10.1016/j.jtte.2020.04.006  

Author Response

Thank you. See the answers in the attachment.

Reviewer 4 Report

Thank you for providing the revised version of your manuscript. Based on the improvement of your work, I suggest publication at this stage.

Author Response

Thank you. See da answers in the attachment.
